# IS DELAYED ROBUSTNESS REALLY GROKKING?

## ABSTRACT

We analyze the phenomenon of delayed robustness, where a neural network trained beyond overfitting becomes robust to adversarial attacks. This phenomenon was first observed by Humayun et al. (2024), and characterized as grokking behavior. We reproduce delayed robustness to PGD attacks in multiple set-ups and, using stronger attacks, show that this robustness is actually overestimated. We then demonstrate that delayed robustness is not grokking, but instead the result of two unintended side effects during overtraining: softmax collapse in the cross-entropy loss function and a too large effective learning rate caused by gradient scaling in the Adam optimizer. We provide experimental evidence that these issues indeed create networks that resist PGD attacks without actually becoming as robust to the stronger attacks. We also point out a relation with dying neurons and the slingshot effect. Using simple interventions to solve these issues, we show that no delayed robustness appears.

## 1 INTRODUCTION AND BACKGROUND

Grokking, where an over-parameterized neural network starts to generalize well past the point of overfitting, has been an active object of deep learning research ever since it was first described by Power et al. (2022). This includes research on relevant training settings, like the importance of weight decay (Liu et al. (2022)) and the design of new optimization techniques to accelerate grokking behavior (Lee et al., 2024). This training past overfitting also uncovers other phenomena that occur late into training, like delayed robustness from Humayun et al. (2024). They show how ReLU networks consistently become robust to adversarial attacks after overtraining. Their hypothesis is that, like grokking, the network undergoes a phase transition that creates robust partitions of input space. We want to further test this hypothesis. Our main goal is to determine whether or not this effect can truly be considered grokking.

In section 2, we discuss what it means for a network to be robust against adversarial attacks and how to accurately determine such robustness. We reproduce and extend the findings of Humayun et al. (2024) and show that these networks are not as robust as they seem. This leads to the alternative hypothesis that delayed robustness can be attributed to unintended side effects caused by overtraining, rather than the networks grokking.

Section 3 describes candidate mechanisms for such side effects. In particular, we look at softmax collapse in the cross-entropy loss degenerating the gradient, and also at issues related to non-convergent behavior of the gradient scaling mechanism in the Adam optimizer. In section 4, we share results of experiments in multiple architectures that, as argued in the subsequent discussion in section 5, show that it is indeed these issues, and not grokking, that underlie the delayed robustness phenomenon.

## 2 PRELIMINARIES: (DELAYED) ROBUSTNESS

We start by explaining adversarial robustness and subsequently reproduce the delayed robustness phenomenon.

## 2.1 WHAT IS A ROBUST NETWORK?

A neural network, like any classifier, should in theory be invariant, or robust, to small and irrelevant changes in the input when making a classification decision. This quality has been a topic in ML research for over two decades (e.g. Dalvi et al., 2004). Formally, a classification neural network $f$ is robust with respect to a sample $x$ with dimension $N$, correctly labeled with label $y$, to a perturbation $\epsilon$ measured by a distance norm $d$ if and only if:

$$\forall x' \in \mathbb{R}^N : |x - x'|_d \le \epsilon \Rightarrow f(x) = f(x'). \tag{1}$$

An adversarial attack is a strategy to modify a sample such that it crosses a decision boundary into a wrong class within such a perturbation. The percentage of input samples still correctly classified after perturbation by such an attack is then a measure of robustness, referred to here as adversarial accuracy but also known as robust accuracy (Croce & Hein, 2020).

## 2.2 DELAYED ROBUSTNESS

To reproduce the results of Humayun et al. (2024), we train an MLP network on 1000 random MNIST dataset (Lecun et al., 1998) samples for a large number of batch iterations with the ADAM optimizer. We have two setups: one uses a mean squared error (MSE) loss function and the other softmax cross-entropy (CE) loss. We then used an $\infty$-norm PGD (Projected Gradient Descent, Madry et al., 2018) attack on the full test set with $\epsilon = 0.06$. Full experimental details are in appendix A.1. From these results, we see a sudden increase in adversarial accuracies long after both train- and test accuracy convergence in both setups. This is shown in figure 1.

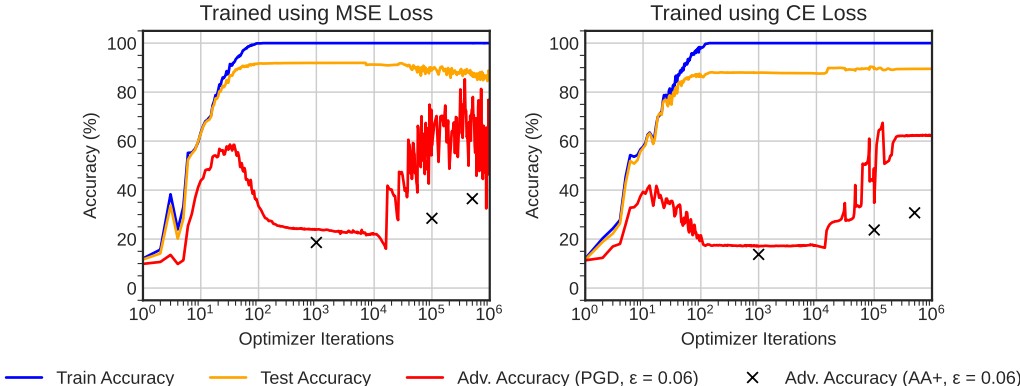

Figure 1: *Train, test and adversarial accuracies (from perturbing the test set using $l_\infty$-PGD with and AA+ $\epsilon = 0.06$) of two MNIST-trained MLPs trained with mean squared error loss (left) and softmax cross-entropy loss (right) respectively.*

Humayun et al. (2024) called this increase *delayed robustness*, and attributed it to the network creating a more robust partition of the input space. In a network with piecewise-linear activations, the network itself is also a piecewise-linear function. So such a network partitions the input space into linear regions, each defined by a unique set of neurons that have positive pre-activations. This concept led to their measure of local complexity: the number of such regions near a data point. They showed a decrease of local complexity around training data points and claimed this was caused by region migration, a phenomenon where these regions move away from the training data and close(r) to the decision boundaries.

However, when we attack these same networks with an attack stronger than PGD, like the parameter-free AutoAttack+ (AA+, Croce & Hein, 2020), the networks show much less adversarial accuracy, which is also illustrated in figure 1. As the amount of perturbation $\epsilon$ increases, the remaining adversarial accuracy decreases; see supplemental figure 9. Since AA+ is also not a perfect attack, this calls into question whether there is even any actual robustness at all, or if the apparent robustness is caused by using too weak adversarial attacks.

## 3 A NEW HYPOTHESIS FOR DELAYED ROBUSTNESS

The results in section 2 show that using PGD, the robustness is overestimated, especially at higher $\epsilon$. Delayed robustness is at least partly a resistance to PGD attacks rather than actual robustness. Since the PGD strategy is to use the gradients of the loss over the input space to maximize the cross-entropy loss (Madry et al., 2018), our hypothesis is that delayed robustness to PGD can be explained by the gradient becoming uninformative for PGD attacks when overtraining. We will explain two different issues that could cause this behavior and also mention solutions that we apply in our experiments.

### 3.1 GRADIENT SCALING IN ADAM

The first issue relates to the Adam optimizer (Kingma, 2014), which uses gradient scaling to adapt the effective learning rate relative to the gradients' magnitudes. At step $t$, a network $f$ with parameters $\theta$ and true labels $y$ over input $x$ is optimized using the following update rules:

$$g_t \leftarrow \nabla_\theta \ell(f(x; \theta_{t-1}), y) \tag{2}$$

$$m_t \leftarrow \beta_1 m_{t-1} + (1 - \beta_1) g_t \tag{3}$$

$$v_t \leftarrow \beta_2 v_{t-1} + (1 - \beta_2) g_t^2 \tag{4}$$

$$\theta_t \leftarrow \theta_{t-1} - \frac{\gamma}{\sqrt{v_t} + \delta} m_t. \tag{5}$$

Here $\gamma$ is the base learning rate and $\delta$ a small value added to avoid division by zero.[1] $\beta_1$ and $\beta_2$ are parameters for the decay rate of the running averages, with default values of $0.9$ and $0.999$ respectively. Note that we omit the bias correction factors $1/(1 - \beta_1^t)$ and $1/(1 - \beta_2^t)$, because in an overtraining regime $t$ is sufficiently large, such that they are practically equal to 1.

When the gradient $g_t$ becomes very small for many steps, so does $v_t$. This causes the scale factor $\frac{\gamma}{\sqrt{v_t} + \delta}$ to increase, and as a result the effective learning rate can become too large. This can cause issues where Adam fails to converge, which has been described before by Reddi et al. (2018). Since $\delta$ acts as an upper bound to the scaling factor, a large enough value of $\delta$ prevents this over-amplification of small gradients. However, in practice, a very small value $\delta = 10^{-8}$ is typically used. As Reddi et al. note: *In practice, selection of the [$\delta$] parameter appears to be critical for the performance of the algorithm.*

One solution is then to increase the $\delta$ term, thereby lowering the upper bound on the effective learning rate. Another solution is used in the AMSGrad optimizer proposed by Reddi et al.. This algorithm keeps track of a maximum for $v$ throughout training, providing an alternative upper bound. This algorithm thus replaces equation 5 by

$$v_t^{\max} \leftarrow \max(v_{t-1}^{\max}, v_{t-1}) \tag{6}$$

$$\theta_t \leftarrow \theta_{t-1} - \frac{\gamma}{\sqrt{v_t^{\max}} + \delta} m_t. \tag{7}$$

### 3.2 SOFTMAX COLLAPSE

A second issue underlying delayed robustness to PGD attacks is softmax collapse, which is caused by numerical precision errors in calculating the softmax cross-entropy loss function or its gradient. The term was introduced by Prieto et al. (2025), who show that the exponential part of cross-entropy loss is at risk of floating-point precision errors, even when written in the more stable form of

$$\ell_{SCE}(\mathbf{z}, y) = -z_y + \max(\mathbf{z}) + \log\left(\sum_{k=1}^n e^{z_k - \max(\mathbf{z})}\right), \tag{8}$$

where $\mathbf{z}$ are the classification network's $n$ output logits, $y$ is the true class label, and so $z_y$ corresponds to the output logit for to the correct class. We discern two types of collapse that both involve

---

[1]Other work often uses $\epsilon$ to denote this small value; here we use $\delta$ to avoid confusion with the perturbation $\epsilon$ from adversarial attacks.

the rightmost term of equation 8. We will refer to this term as the `logsumexp` term, after the equivalent PyTorch function[2].

UNDERFLOW SOFTMAX COLLAPSE

Underflow softmax collapse occurs when the difference between an incorrect logit and the maximum logit becomes so large that $e^{z_k - \max(\mathbf{z})}$ does not fit within the numerical range of the float representation anymore and evaluates to 0 (underflow). If none of the incorrect logit differences fit anymore, only the term from the correct logit $z_y = \max(\mathbf{z})$ is left, and the whole `logsumexp` returns 0. Because

$$\frac{\partial \ell_{SCE}}{\partial z_c} = \frac{e^{z_c - \max(\mathbf{z})}}{\sum_{k=1}^{n} e^{z_k - \max(\mathbf{z})}} - \mathbf{1}_{c=y}, \tag{9}$$

the gradients with respect to underflowed logits also underflow, while increasing the gradients with respect to the remaining logits. This phenomenon was also called gradient vanishing, and first described by Carlini & Wagner (2017).

ABSORPTION SOFTMAX COLLAPSE

A second type of softmax collapse, absorption softmax collapse, was described by Prieto et al. (2025). Absorption occurs when adding a small float to a much larger float, and the result is rounded to the largest float. We refer to Prieto et al. (2025) for a more detailed description. They note that the `logsumexp` term can also reduce to 0 for a correctly predicted sample through such floating-point absorption,

$$\sum_{k=1}^{n} e^{z_k - \max(\mathbf{z})} \doteq e^0 = 1, \tag{10}$$

where $\doteq$ denotes floating-point equality.

This type of softmax collapse can still occur even when all the exponential terms fit within float precision individually. Also, the gradient of this type of softmax collapse is only 0 for the correct class logit, while the gradients of others become equivalent to their exponential term, as demonstrated by combining equation 9 and equation 10.

SOLUTIONS

A band-aid solution to both types of softmax collapse is to simply increase the numerical precision of the calculations. However, any finite-precision calculation is a temporary solution, since logits keep growing using cross-entropy loss through a process called naive loss minimization (Prieto et al., 2025). Note that for absorption softmax collapse, it is sufficient to increase the precision only when calculating the loss and the logit gradients from the network's output logits, before back-propagation. For underflow softmax collapse, it is also necessary to increase the precision of the back-propagation.

Another solution is to scale the logits to an input domain where the calculation is numerically stable. This results in a scaled variant of softmax cross-entropy loss,

$$\ell_{ScaledSCE}(\mathbf{z}) = \ell_{SCE}(s(\mathbf{z})\mathbf{z}), \tag{11}$$

$$s(\mathbf{z}) = \min\left(1, \frac{s_{\max}}{\max_{j,k}(z_j - z_k)}\right), \tag{12}$$

where $s_{\max} = 20$ is the maximum allowed difference between logits. This loss function can be used as a drop-in replacement for cross-entropy loss in a PGD attack. This rescale-and-attack strategy, which we call ScaledPGD, can be used to adversarially attack samples where normal PGD fails because of softmax collapse.

---

[2]https://docs.pytorch.org/docs/stable/generated/torch.logsumexp.html

# 4 EXPERIMENTS

We perform a number of experiments to show that Adam convergence issues and softmax collapse underlie delayed robustness to PGD. We perform most of these experiments with MNIST-trained MLPs, but we also validate our results on a CIFAR10-trained ResNet18. We refer to appendix A.1 for further experimental details.

## 4.1 MEAN SQUARED ERROR

For our first experiments, we train a four-layer MLP with the same initialization on the same MNIST subset of 1000 samples for $10^7$ optimizer iterations with three different optimizers. Two runs use the Adam optimizer, one with the default $\delta$ value of $10^{-8}$ and another with a value $\delta = 10^{-4}$. The third run uses the AMSGrad optimizer with the default $\delta = 10^{-8}$.

For each run, we track the train and test accuracy as well as the train set loss and the adversarial accuracy on the test set using PGD. More details can be found in the appendix A.1, and the results are in figure 2. Overtraining using Adam with default parameters and mean squared error loss shows delayed robustness to PGD, but also deterioration of the network's accuracy and a very unstable loss. The two other optimizers clearly stabilize the loss and also do not cause delayed robustness to PGD.

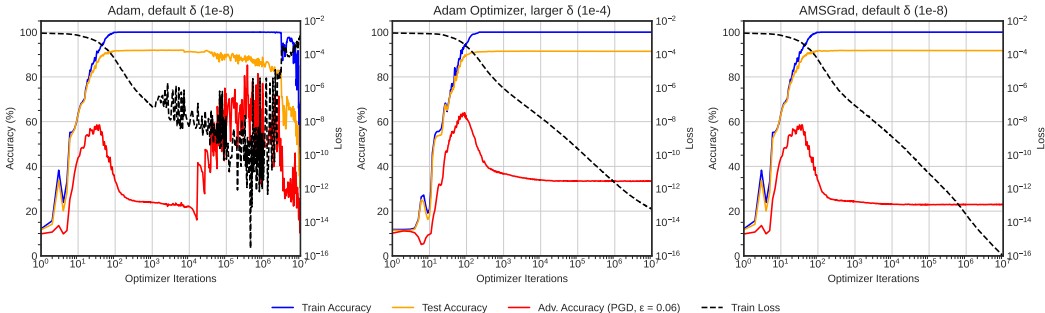

Figure 2: *Train, test, and adversarial accuracies (from perturbing the test set using $l_\infty$-PGD with $\epsilon = 0.06$), as well as train set loss of three MNIST-trained MLPs trained with mean squared error loss. One uses the default Adam optimizer (left), another Adam with a higher $\delta = 10^-4$ (middle) and the last one uses the AMSGrad optimizer (right).*

To understand why the first network ends up in such a poorly performing state, we analyzed both the number of dead neurons as well as the local complexity, calculated as per Humayun et al. (2024). We consider a ReLU neuron dead if it has 0 post-activation for the entire training data set. The results can be found in figures 3 and 4. They show that using Adam with $\delta = 10^{-8}$, the local complexity starts decreasing for training samples, very similar to Humayun et al.'s figure 2, around the same time that the loss becomes unstable. Subsequently, dead neurons start accumulating throughout the layers. In the network trained with AMSGrad, however, the local complexity for the train samples rises smoothly before evening out, while the number of dead neurons remains stable.

## 4.2 CROSS-ENTROPY

Our next experiments also use a four-layer MLP, but this time trained using cross-entropy loss for 1 000 000 optimizer iterations. We train and attack the network twice, once using 32-bit precision like before, and once using 64-bit precision. We track the same metrics as for the mean squared error networks, as well as the adversarial accuracy obtained by ScaledPGD. The results are in figure 5.

In 32 bits, we again see an unstable training loss. However, this time these instabilities did not affect the train accuracy and instead marginally improved our test accuracy. Using ScaledPGD, we obtain a far lower adversarial accuracy, just like using AA+ did in figure 1. In supplemental figure 10 we also show the difference between default PGD and ScaledPGD attack accuracies at different perturbation

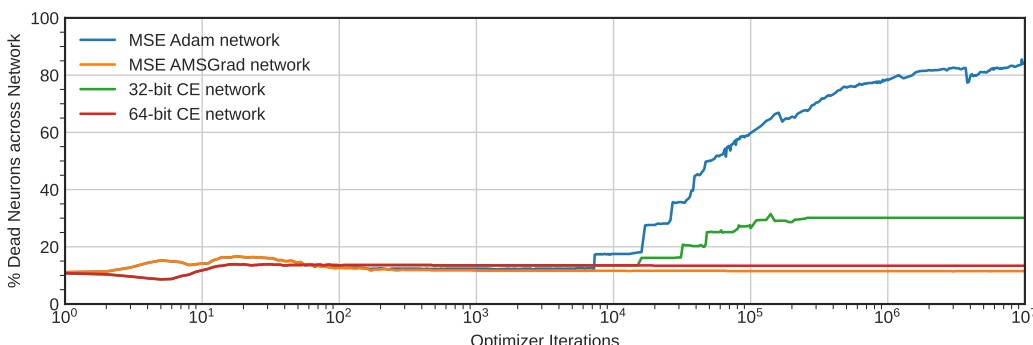

Figure 3: *Analysis of the number of dead neurons throughout training of the MLPs on 1000 MNIST samples in our experimental set-ups.*

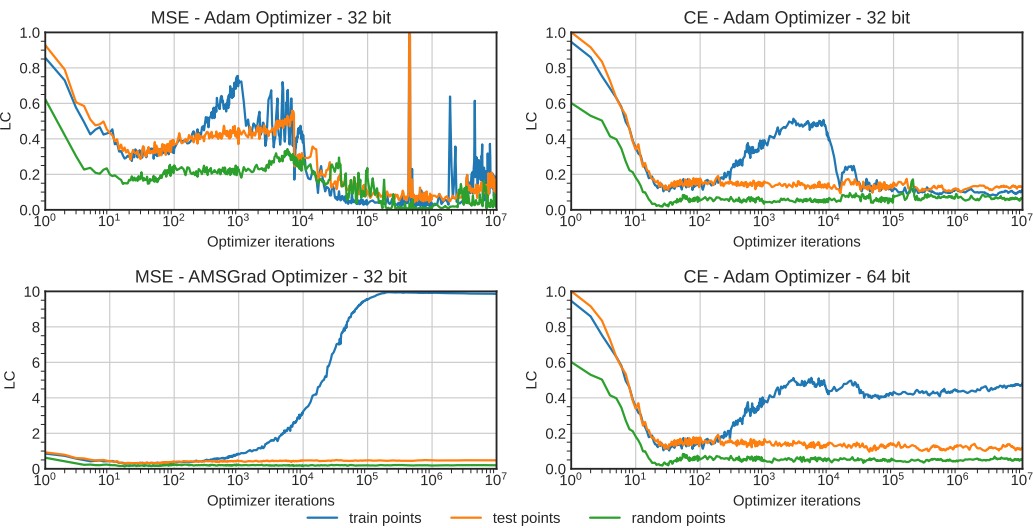

Figure 4: *Local Complexity (LC), around train, test, and random data points, throughout training of the MLPs on 1000 MNIST samples.*

levels of this network. When training with 64-bit precision, delayed robustness is completely absent and all metrics are stable.

To show that it is softmax collapse underlying these instabilities in the loss, we count the occurrences of softmax collapse during training. These results can be found in figure 6, which shows that in the 32-bit network there is a high amount of softmax collapse when the loss instabilities occur. With 64 bits there is only a steadily increasing amount of absorption softmax collapse.

In figures 3 and 4 we show the number of dead neurons and the local complexity in these experiments. With 32-bit precision, there are sharp increases in the number of dead neurons, co-occurring with the loss instabilities. This is preceded by a sharp drop in local complexity around training samples. In the 64-bit network, both the local complexity and the number of dead neurons are stable.

### 4.3 CIFAR10 RESNET18 VALIDATION

To show that our results generalize to different architectures and datasets, we also perform experiments on a ResNet18 (He et al., 2016) architecture. We trained the network on the the full 50 000 sample CIFAR10 dataset (Krizhevsky et al., 2009) for 1 000 000 optimizer iterations. Like Humayun et al. (2024) we do not use batch normalization. We again evaluate the difference between the de-

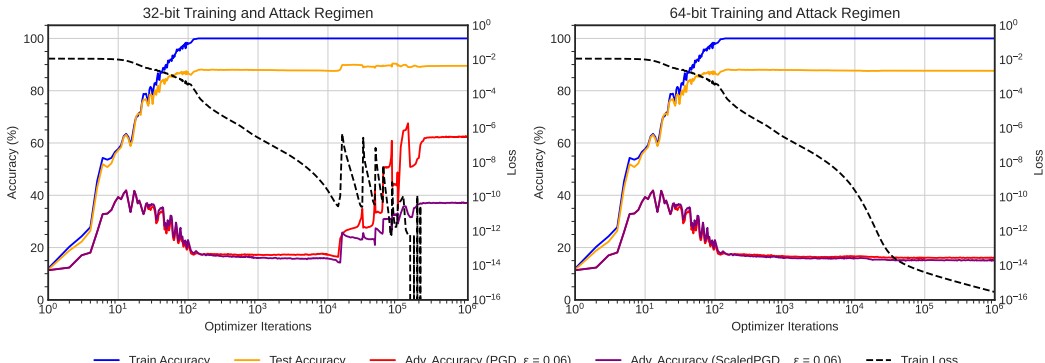

Figure 5: *Train, test, and adversarial accuracies (from perturbing the test set using $l_\infty$-PGD at both 32- and 64-bit precision and $l_\infty$-ScaledPGD) with $\epsilon = 0.06$), as well as train set loss of two MNIST-trained MLPs trained with softmax cross-entropy loss. One was trained using default 32-bit loss calculation (left), the other used a 64-bit loss calculation.*

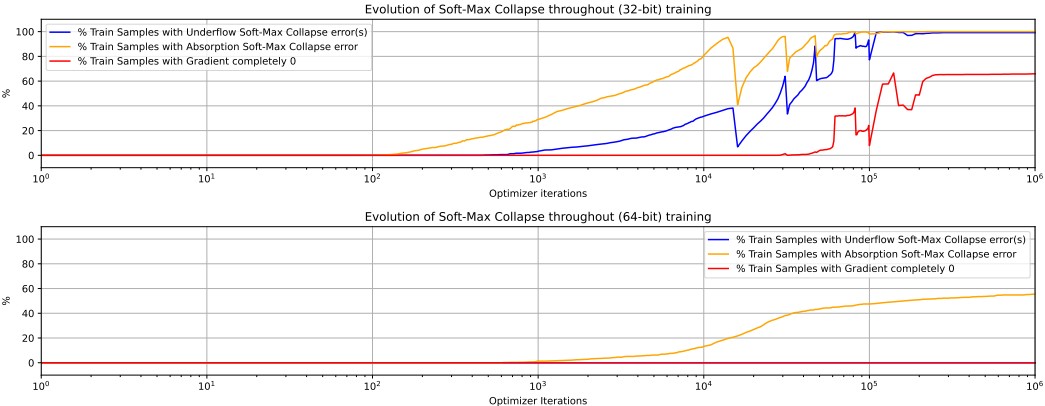

Figure 6: *Softmax collapse errors in the output logit gradients throughout training of the MLPs on 1000 MNIST samples using 32 bits (top) or 64 bits (bottom) precision.*

fault Adam optimizer and the AMSGrad optimizer when using mean squared error loss, and the difference between 32- and 64-bit training when using cross-entropy loss. For these experiments, we report adversarial accuracies with respect to PGD, as well as AA+'s Auto-PGD attack (for mean squared error loss) and ScaledPGD (for cross-entropy loss). We use only the AA+'s Auto-PGD attack, described in Croce & Hein (2020), instead of the full AA+, because it already reduces the adversarial accuracy to nearly 0. All these attacks were done with $\epsilon = 0.06$. The results are in figure 7. Plots with more metrics are in the appendix (figures 11 and 12).

The adversarial accuracy results show delayed robustness to PGD in all the experiments. The mean squared error network trained with Adam shows a small and noisy gain in adversarial accuracy when attacked by PGD. Using AMSGrad further decreased and stabilized this adversarial accuracy. The cross-entropy models' results instead show large sudden gains in adversarial accuracy to PGD. This closely resembles the pattern of the ResNet experiments of Humayun et al. (2024). Increasing the precision from 32 to 64 bits only delayed this pattern, rather than fully preventing it. We show a similar delay in the development of softmax collapse errors in figure 8. Contrary to all these cases, the Auto-PGD and ScaledPGD attacks leave practically no adversarial accuracy.

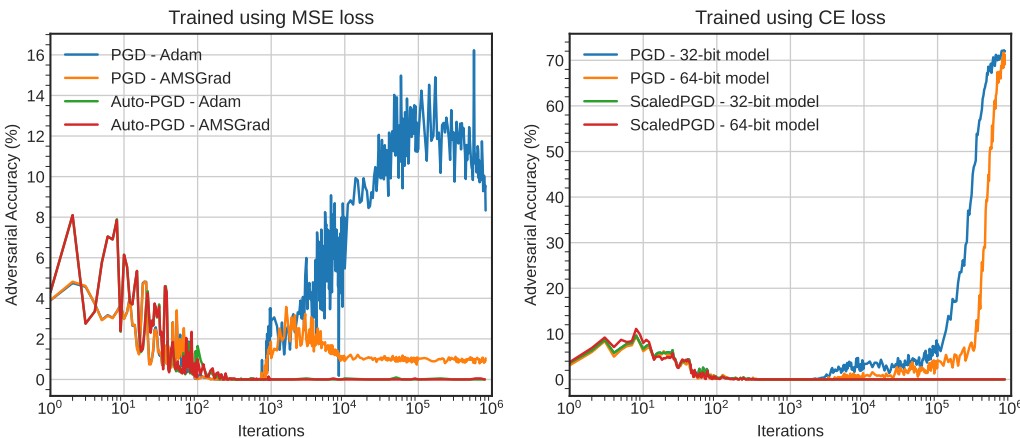

Figure 7: *Adversarial accuracies in the different ResNet18 setups.* $\epsilon = 0.06$.

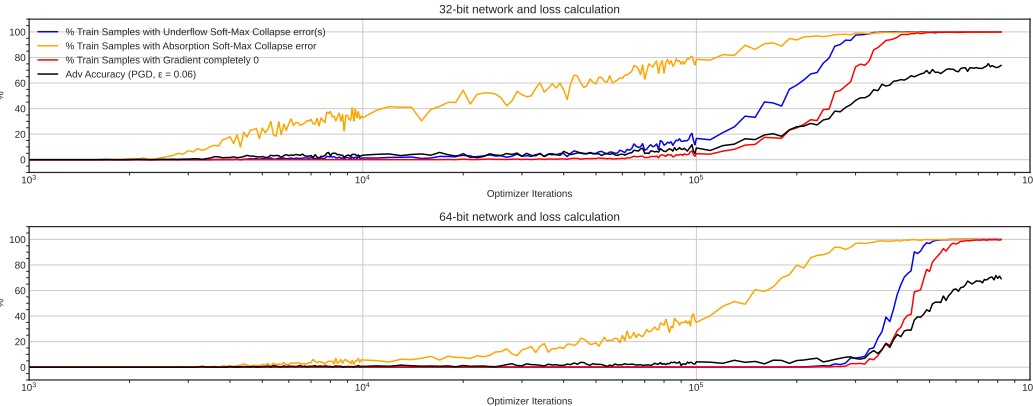

Figure 8: *Softmax collapse errors in the output logit gradients throughout training of the ResNet on CIFAR10 using 32-bit (top) or 64-bit (bottom) precision. Also includes the adversarial accuracy against PGD attacks on the test set at $\epsilon = 0.06$. Please note the x-scale starting at $10^3$ iterations.*

## 5  DISCUSSION

The theoretical and experimental evidence presented shows that the delayed robustness phenomenon reported by Humayun et al. (2024) is caused by unintended side effects during overtraining. Using mean squared error loss and the Adam optimizer, we managed to precisely reproduce their results on MNIST. There saw delayed robustness against PGD attacks, but also observed an unstable loss and an accumulation of dead neurons, which ultimately harmed the network's performance. Our results indicate that all of these effects are due to Adam's gradient scaling causing a too high effective learning rate. Using AMSGrad prevented the loss instabilities, stopped neurons from dying, and stabilized local complexity.

In the 32-bit MNIST cross-entropy experiments, we also saw delayed robustness using PGD and even some gain in adversarial accuracy using ScaledPGD. These gains were also associated with loss instabilities, a drop in local complexity, and dying neurons. Here, these effects co-occur at some critical threshold of softmax collapse. We show that these effects happen consistently in supplemental figures 16 and 17, where we repeat these findings using different random subsets of 1000 MNIST samples and other model initializations. In our supplementary experiments, we noted that using LeakyReLU does not alleviate the neurons dying (see supplemental figures 13, 14, and 15), as they can still get into a permanently negative output state. By increasing the model's

precision to 64 bits, we stopped the majority of collapse and thereby also all the subsequent effects, which confirms that numerical errors were their cause.

The loss instabilities of the cross-entropy experiments are reminiscent of the slingshot effect from grokking literature, which Thilak et al. (2022) described as a non-monotonic training loss behavior characterized by sudden large spikes. These are especially clear when we plot the train set loss with a linear scale for the iterations, which we do in supplemental figure 18. Doing the same for mean squared error losses in supplemental figure 19, we also see slingshot-like behavior. Prieto et al. (2025) mention in their appendix H that the slingshot effect is a known side effect of adaptive gradient methods and discuss its relationship with grokking and softmax collapse. They state that while the slingshot effect can induce grokking, it is not the same as, nor a requirement for, grokking. This was shown by Nanda et al. (2023), who write in their own appendix (section D.2) that they suspect that slingshots *"serve as an implicit regularization mechanism that favors the simpler, generalizing solution over the more complicated"*. This is in line with our findings that the loss instabilities co-occur with dead neurons, which cause a lower complexity of the remaining network.

We also showed that, contrary to what was claimed by Humayun et al. (2024), ResNets are not actually becoming robust when overtrained. In ResNets trained on CIFAR10 using mean squared error and Adam, AA+'s Auto-PGD attack was still fully effective. Auto-PGD uses an adaptive step size while attacking (Croce & Hein, 2020), which leads us to conclude that the PGD attack occasionally ends up in local loss maxima because of its fixed step size. ResNets trained using cross-entropy loss showed delayed robustness to PGD, but were still fully vulnerable to ScaledPGD. This delayed robustness can therefore be attributed to softmax collapse errors.

For ResNets trained with cross-entropy loss, increasing the precision did not fully remove the delayed robustness to PGD, but only created a further delay. This is because, on top of naive loss minimization, skip connections cause the output variance of ResNets to scale exponentially with depth (Zhang et al. (2019)). The logits thus scale so fast that they reach the limits of both 32- and 64-bit numerical precision within a million updates, as shown in figure 8. Batch or layer normalization is normally used to resolve this, as they both normalize the intermediate values after each skip connection. In fact, Humayun et al. already showed an absence of delayed robustness to PGD when batch normalization was used in their figure 12.

Altogether we have shown that delayed robustness is primarily an artifact of weak adversarial attacks, numerical errors, and training loss instability. The delayed robustness mostly disappears with more stable training methods and stronger attacks. The small amount of delayed robustness that remains in our MNIST experiments is likely caused by the slingshot effect. We hypothesize that the slingshot mechanism moves the network's parameters to different local loss optima that have fewer effective parameters after accounting for dead neurons. Future work could determine if this is the case and if so, whether this behavior can be exploited to benefit training, generalization, or grokking in a controlled way, like Thilak et al. also ask for the slingshot effect in general. Our findings, where the slingshots seem to be induced by softmax collapse and cause ReLU neurons to die, may provide new clues for how to do so. Further efforts could also go into the design, evaluation, and adoption of stable versions of widely used training algorithms. This is especially important if overtraining is going to be the new norm. We already showcased the benefits of AMSGrad. We also want to point future research to Prieto et al. (2025), who proposed StableMax as a collapse-free softmax alternative and the $\perp$Grad optimizer that prevents naive loss minimization.

## REPRODUCIBILITY STATEMENT

See the appendix A.1 for the most important experimental details and also our repository for code to reproduce our main results.[3]

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

# A  APPENDIX

## A.1  EXPERIMENTAL DETAILS

The experiments described in this paper were all performed using PyTorch [4] under the most basic settings possible. As an optimizer, we used Adam unless specified it was AMSGrad[5], always

---

[4] https://pytorch.org/

[5] In Pytorch this is still the Adam optimizer but with the setting `amsgrad = True`.

with the default learning rate of 0.001 and the default values for other parameters unless explicitly specified otherwise.

We stored and analyzed our models on base-10 logarithmic intervals during training. We used a manual torch seed of 42 wherever possible before initializing the networks, with the exception of the repeat experiments mentioned in the discussion (section 5), where we used different seeds.

The MLP networks in were all 4-layer ReLU MLPs with a layer width of 200. The indices of the 1000 random MNIST samples were always the same except in the repeat experiments mentioned in the Discussion, and can also be found on our project repository.

The ResNet18 models from section 4.3 were as similar as possible as those from Humayun et al. (2024). The model-generating script we both used can be found in our repository. Like them we did not use batch-normalization and used a width-parameter of 16.

For the adversarial attacks, we borrowed the PGD code from Humayun et al. (2024) and it can be found on our repo together with code for the ScaledPGD attack. For all attacks we used the $\epsilon$-perturbations as described in the main text (default 0.06), with a step size of 0.0156 and 10 PGD steps. We did limit the attacks to the [0, 1]-domain. Auto-Attack, that we used in section 2.2, is parameter-free. The only relevant settings we used for that were `norm='Linf'`, `eps=e`, `version='plus'` and the fact that we restricted it to only use `'apgd-ce'` in some experiments.

We also used code from Humayun et al. (2024) for our own local complexity results. What we used can be found in the repository as well.

## A.2 SUPPLEMENTAL FIGURES

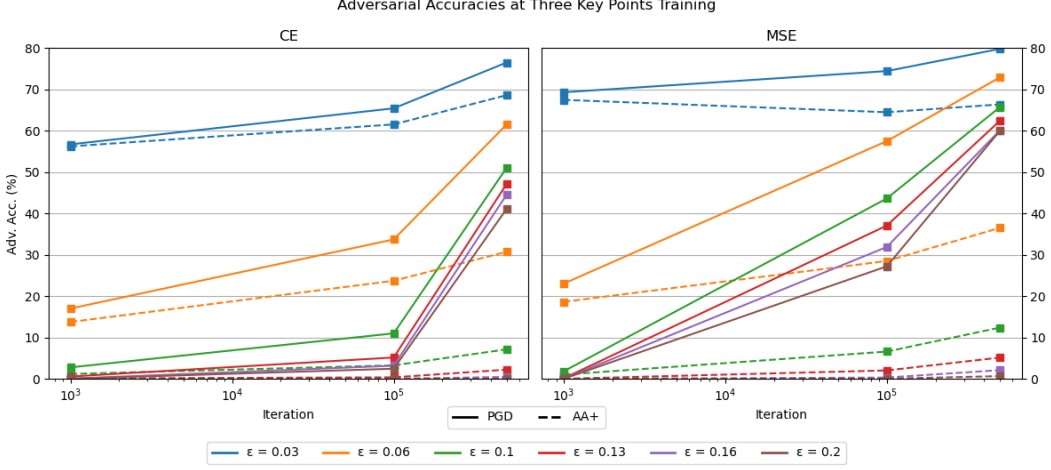

Figure 9: *Adversarial Accuracies when attacking the cross-entropy (left) and mean squared error (right) networks from figure 1 with both Auto-Attack+ and PGD at three different times during training (at iterations $10^3$, $10^5$ and $5 \cdot 10^5$).*

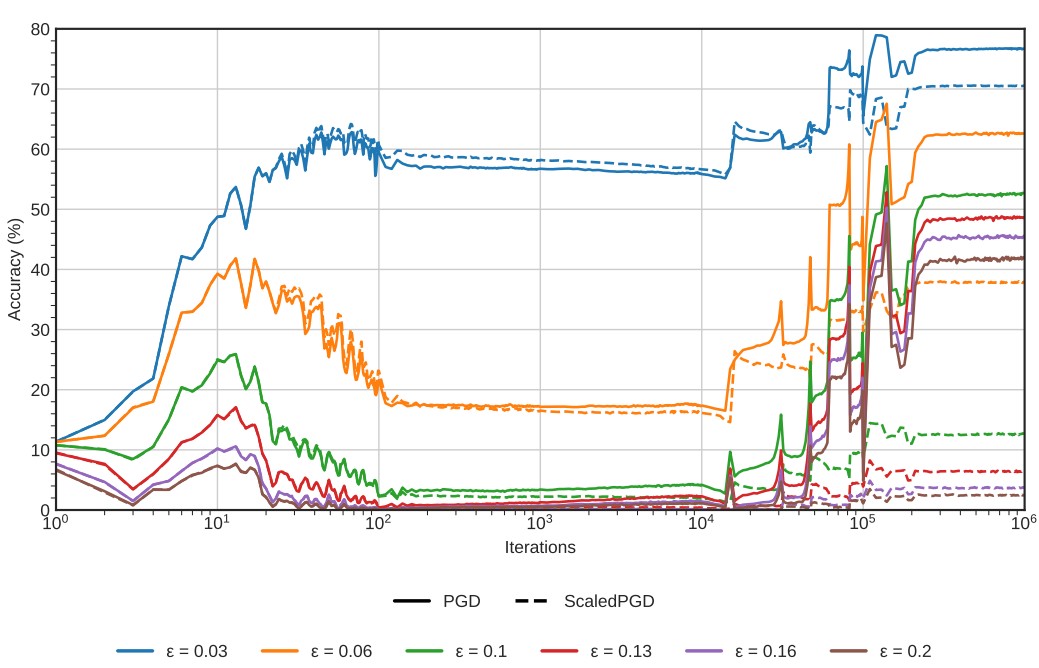

Figure 10: *Comparison of adversarial accuracies at different $\epsilon$ pertubation values for two attacks: $l_\infty$-PGD and $l_\infty$-ScaledPGD.*

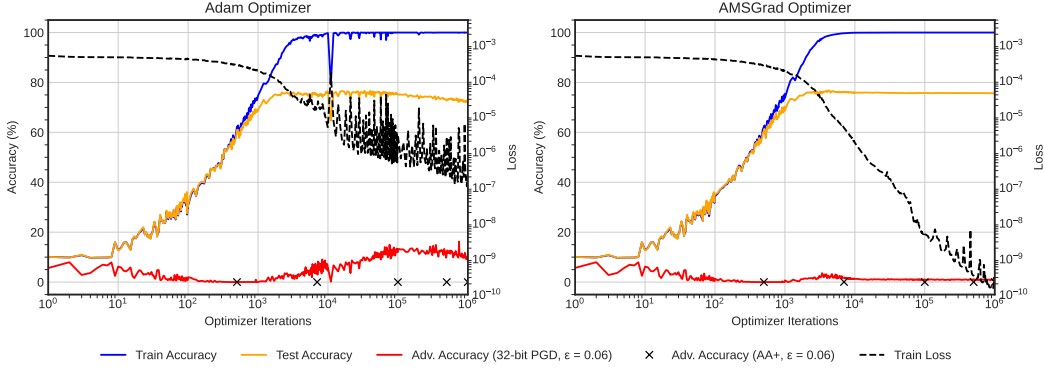

Figure 11: *Train, test and adversarial accuracies, from perturbing the test set using $l_\infty$-PGD with $\epsilon = 0.06$, as well as train set loss of two CIFAR10-trained ResNets with mean squared error loss. One uses default Adam optimizer (left), the other uses the AMSGrad optimizer (right).*

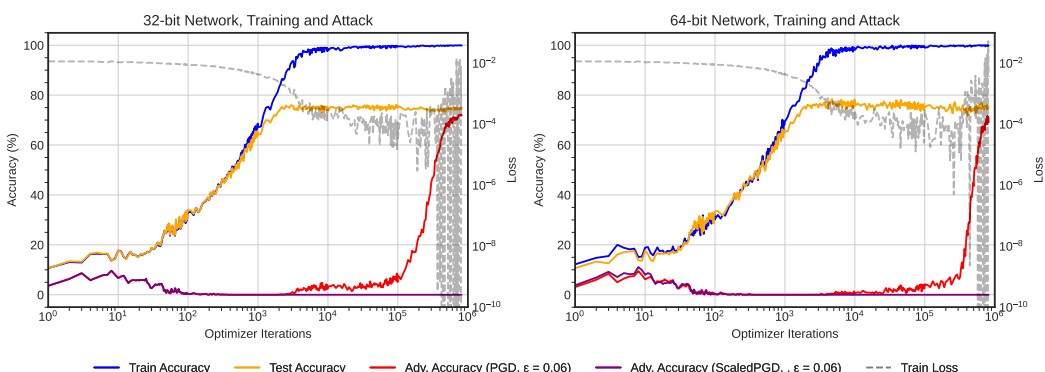

Figure 12: *Train, test and adversarial accuracies (from perturbing the test set using $l_\infty$-PGD and $l_\infty$-ScaledPGD with $\epsilon = 0.06$), as well as train set loss of two ResNets trained on CIFAR10 with softmax cross-entropy loss. One used a 32 bits regimen (left), the other 64 bits.*

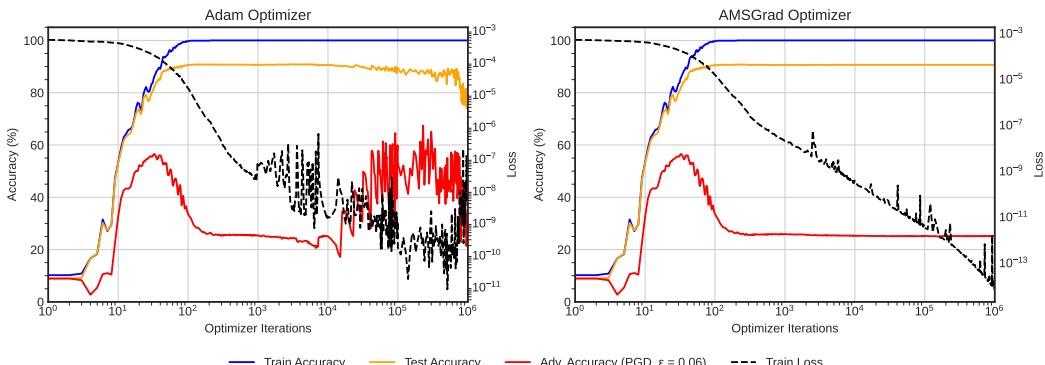

Figure 13: *Train, test and adversarial accuracies (from perturbing the test set using $l_\infty$-PGD with $\epsilon = 0.06$), as well as train set loss of two MNIST-trained MLPs trained with mean squared error loss and **leakyReLU activation functions**. One uses the default Adam optimizer (left) the other the AMSGrad optimizer (right).*

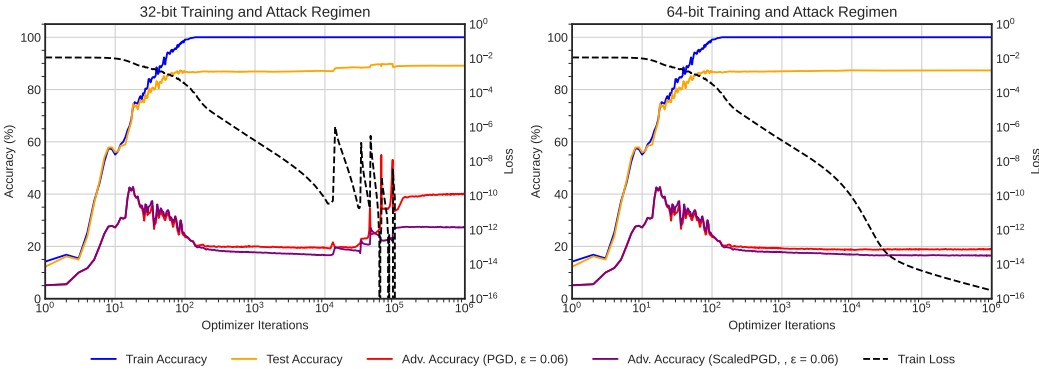

Figure 14: *Train, test and adversarial accuracies (from perturbing the test set using $l_\infty$-PGD and $l_\infty$-ScaledPGD) with $\epsilon = 0.06$), as well as train set loss of two MNIST-trained MLPs trained with softmax cross-entropy loss and **leakyReLU activation functions**. One was trained using default 32-bit loss calculation (left), the other used a 64-bit loss calculation.*

702
703
704
705
706
707
708
709
710
711
712
713
714
715
716
717
718
719
720
721
722
723
724
725
726
727
728
729
730
731
732
733
734
735
736
737
738
739
740
741
742
743
744
745
746
747
748
749
750
751
752
753
754
755

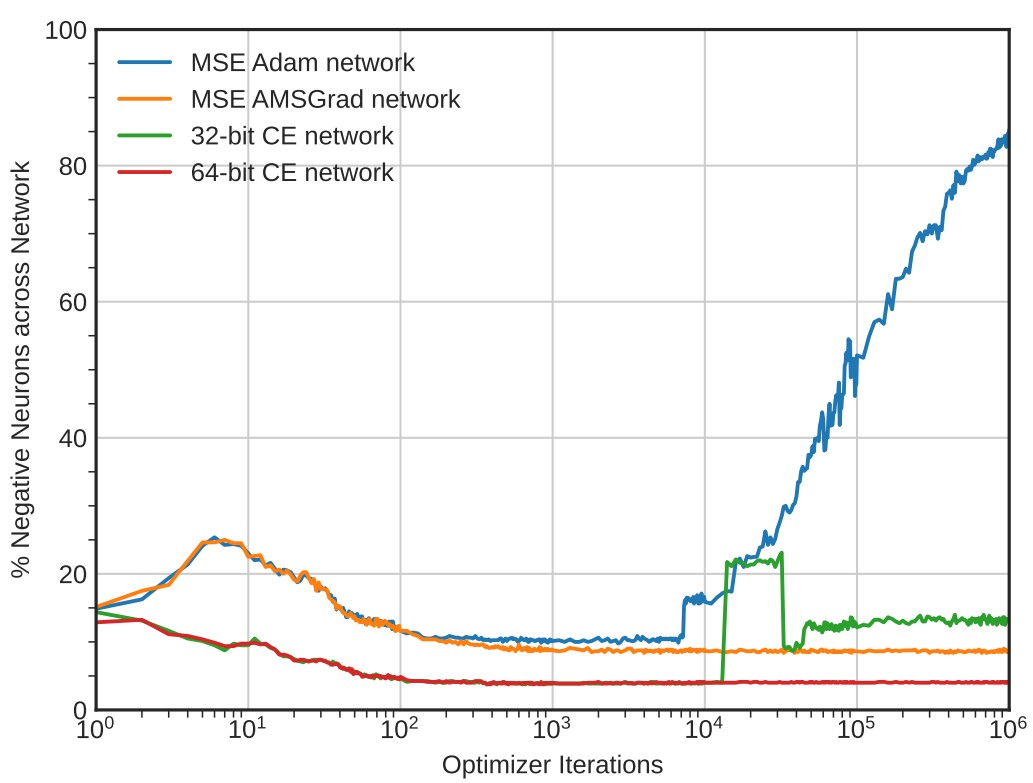

Figure 15: *Analysis of the number of fully negative neurons throughout training of the MLPs **using leakyReLU activation functions** on 1000 MNIST samples in our experimental set-ups.*

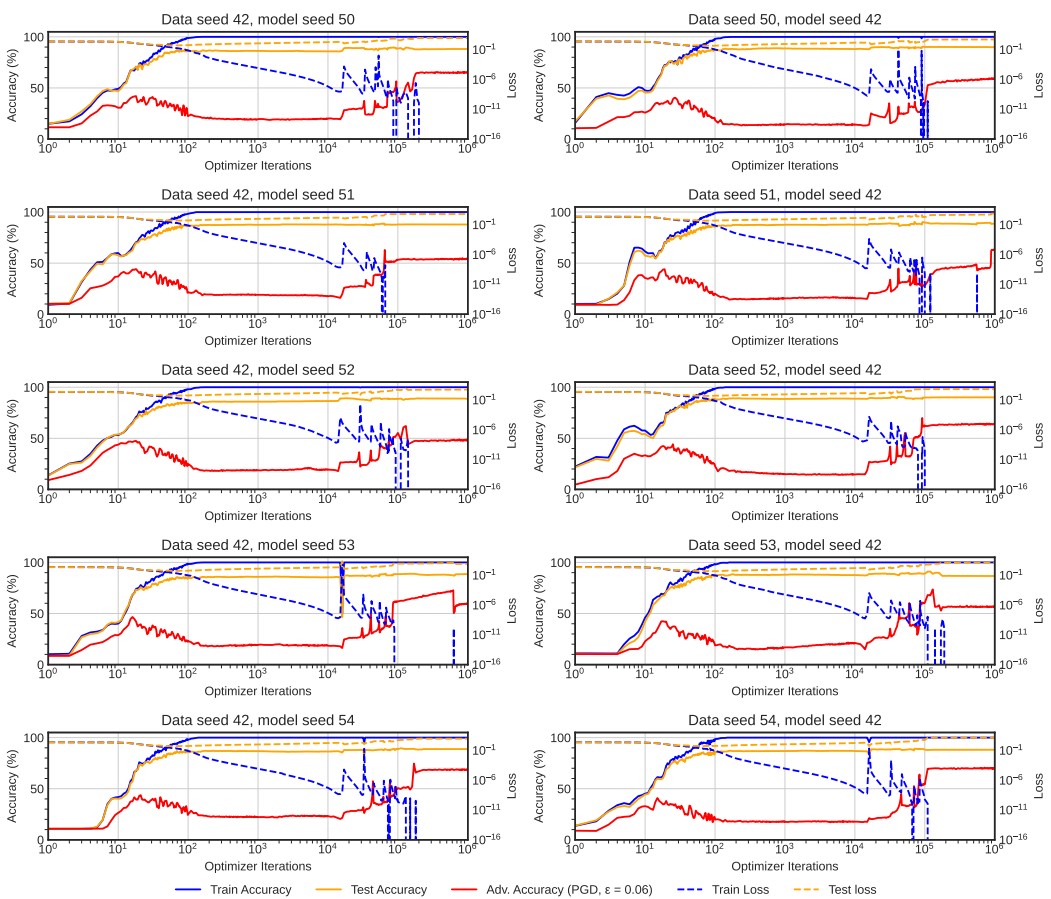

Figure 16: *Accuracy and loss results for 10 repeat runs with different sampling- and model seeds for the MNIST MLP experiments.*

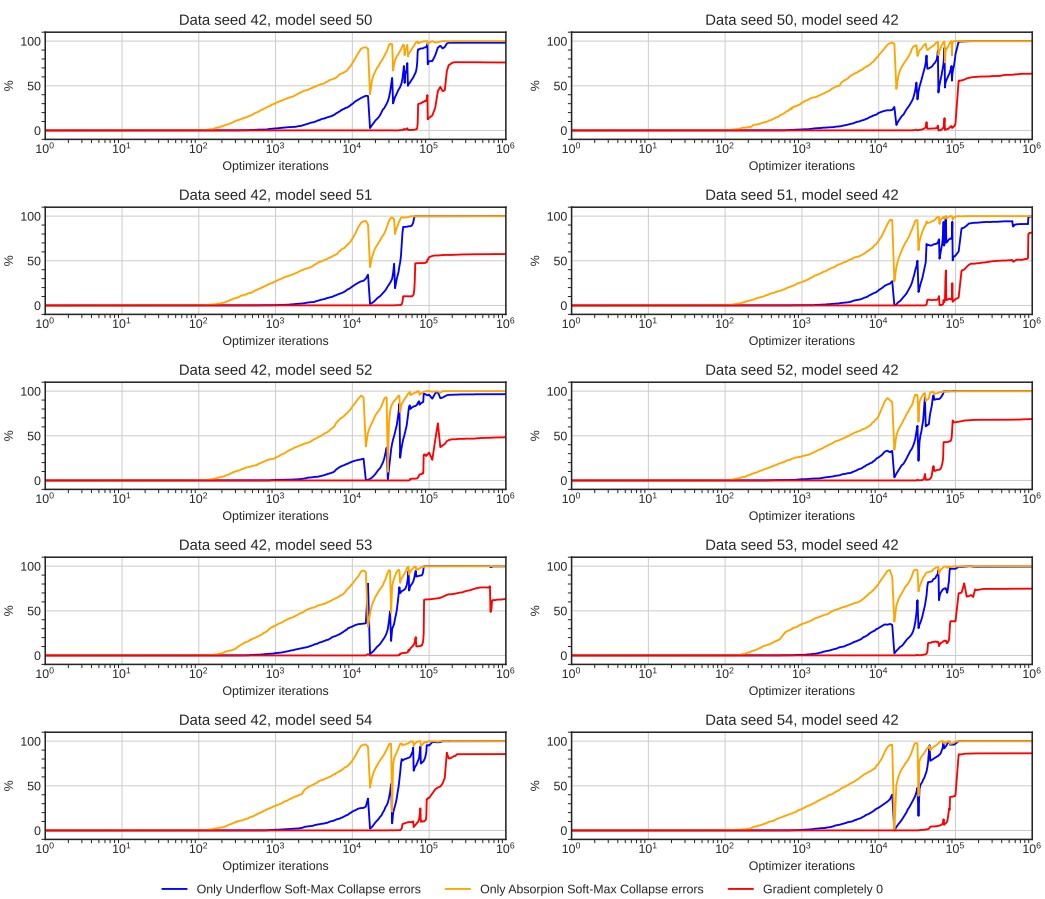

Figure 17: *Softmax collapse results for 10 repeat runs with different sampling- and model seeds for the MNIST MLP experiments.*

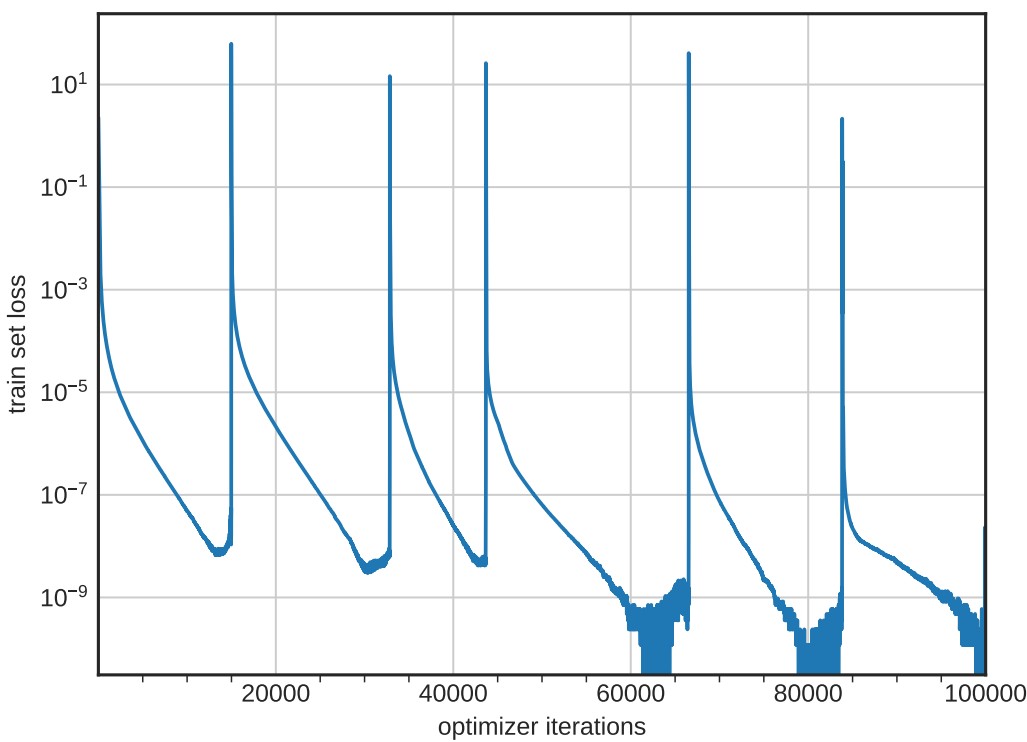

Figure 18: *The train set **cross-entropy** loss of the first 100k iterations of a repeat experiment similar to figure 16 (model seed 42, data seed 42) to show slingshot effect-like behavior. Contrary to other plots in this paper, these were recorded at every iteration and the x-axis is **not** log-scaled.*

918
919
920
921
922
923
924
925
926
927
928
929
930
931
932
933
934
935
936
937
938
939
940
941
942
943
944
945
946
947
948
949
950
951
952
953
954
955
956
957
958
959
960
961
962
963
964
965
966
967
968
969
970
971

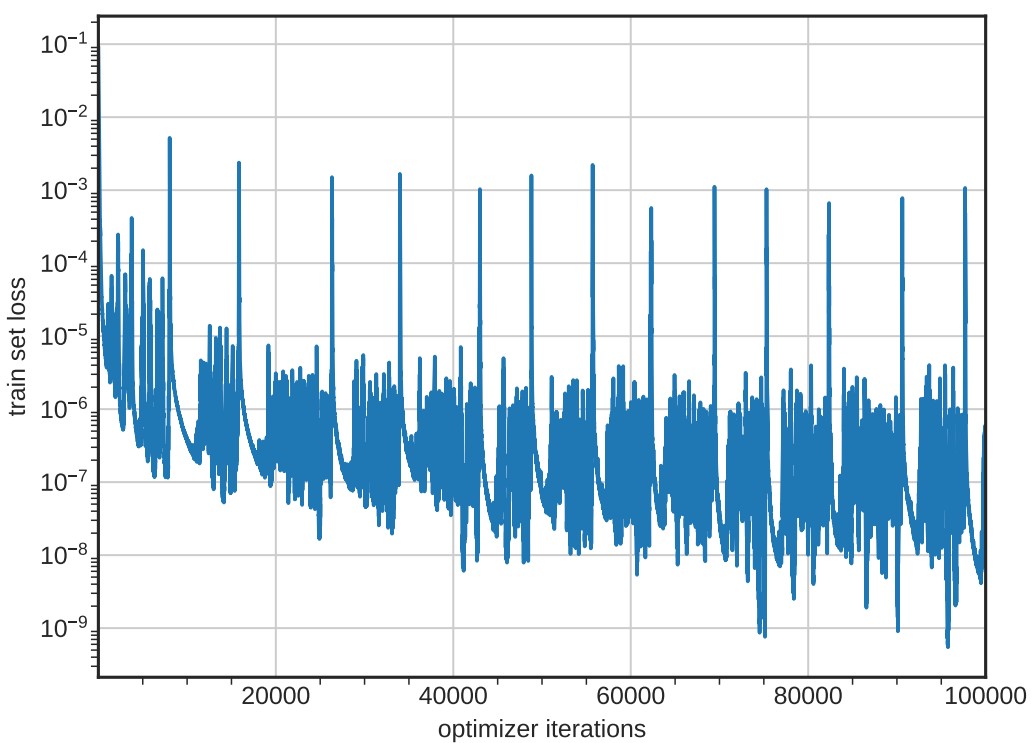

Figure 19: *The train set **mean squared error** loss of the first 100k iterations of a repeat experiment similar to figure 16 (model seed 42, data seed 42) to properly show slinghot effect-like behavior. Contrary to other plots in this paper, these were recorded at every iteration and the x-axis is **not** log-scaled.*

