# OpenReview forum: "Is Delayed Robustness Really Grokking?"
_ICLR.cc/2026/Conference — Submitted to ICLR 2026_

### Official Review · Reviewer_sBXh · 2025-10-28

**Soundness:** 2
**Presentation:** 3
**Contribution:** 2
**Rating:** 4
**Confidence:** 3

**Summary:**

This paper offers a critical re-evaluation of the "delayed robustness" phenomenon, previously equated with Grokking, demonstrating compellingly that the observed robustness is significantly overestimated. The paper analyzes the causes from two perspectives—Softmax collapse and the Adam optimizer—and subsequently eliminates delayed robustness through gradient clipping and optimizer manipulations.

**Strengths:**

1. This paper intuitively validates through experimental observation that delayed robustness is susceptible to more powerful adversarial attacks, and the phenomenon is made to vanish by modifications addressing both Softmax collapse and the optimizer.
2. The paper is generally well-structured and easy to follow. The arguments are presented with adequate clarity.

**Weaknesses:**

1. Insufficient discussion of the deep mechanism of the Grokking phenomenon. The explanation lacks a formal theoretical framework or analytical evidence linking these mechanisms to the observed robustness behavior.
2. The selection and comparison of adversarial attack methods could be further enhanced. While the paper uses AutoAttack+ to demonstrate the inadequacy of PGD, it may not have fully explored other advanced black-box or white-box attack methods.

**Questions:**

1. The connection between these numerical effects and the phase-transition characteristics typically associated with grokking is not clearly established. Additional theoretical modeling or quantitative diagnostics would be necessary to strengthen the argument.
2. The authors might consider providing experimental results covering more adversarial attack methods. This expanded evaluation would help ensure the analysis is more comprehensive.

---

> ### Author Response · Authors · 2025-11-28
>
> Dear reviewer,
>
> We thank you for taking your time to review our work.
>
> We would like to address your questions/weaknesses simultaneously:
>
> 1. Indeed, we do not go deep into the mechanism of grokking. This is mostly because we believe we show that delayed robustness is independent of grokking. They are both different phenomena that happen during overtraining. The only real relation we find is in the slingshot effect causing dead neurons, which we propose in the Discussion section. We therefore do not give a general explanation of the mechanism behind grokking, except from showing that delayed robustness is *not* part of such an explanation. We will make this more clear in the text.
>
> 2. We argue that, since AutoAttack+ is already an ensemble of multiple parameter-free attacks, it is state-of-the-art for our purpose. Our goal was to find any attack that falsifies delayed robustness, especially at higher perturbation levels, which AutoAttack+ does.

---

### Official Review · Reviewer_8wjS · 2025-10-31

**Soundness:** 3
**Presentation:** 2
**Contribution:** 2
**Rating:** 2
**Confidence:** 4

**Summary:**

This paper studies a phenomenon called delayed robustness, where an over-trained neural network becomes robust to adversarial attacks.
This phenomenon was introduced by Humayun et al. (2024).

This paper presents evidences that delayed robustness is an artifact of improper evaluation.
Specifically, the authors show that the true robustness does not increase for over-trained under stronger attacks.
In addition, the authors also present alternative hypothesis for why delayed robustness was observed in the first place.

**Strengths:**

1. This paper refutes the claim by Humayun et al. (2024).
Thus, publishing this paper could potentially fix incorrect statements in the existing literature.
1. Overall, I think the claims in this paper makes sense.
In the past, there were a lot of work in the adversarial robustness literature that turned out to be ineffective (or less effective) under stronger attacks.
And it seems that delayed robustness also falls into this category.

**Weaknesses:**

1. Experiments are only done on MNIST and CIFAR10.
Ideally, it would to great to validate the claims in the paper on larger datasets like ImageNet.
1. The impact of this paper is somewhat limited.
This paper seems to be written in a way that specifically refutes the delay robustness claim by Humayun et al. (2024).
1. There are a lot of stuff in the experiments with different settings, and it would be better to consolidate a bit.
For example, a better way to present the results is to follow the same training procedure (with 32-bit floating point numbers, cross entropy loss, and Adam) in Humayun et al. (2024), and show that delayed robustness does not emerge under stronger attacks (e.g., auto-PGD and PGD with scaled softmax.)

Minor.
1. I am assuming experiments in Section 4.1 are trained with the mean squared error loss.
1. Typo in line 473 "\\(\perp\\)Grad"?
It would be better to explicitly mention this in the text, instead of letting readers infer it from the section title.

**Questions:**

1. Why do some experiments use the MSE loss to train the neural network even though MNIST and CIFAR10 are classification datasets?
The use of MSE as a training loss does not seem to relate to any of the hypothesis mentioned in Section 3.
Note that the cross entropy loss, as described in Section 4.2, affects PGD attack in the test time.

---

> ### Author Response · Authors · 2025-11-28
>
> Dear reviewer,
>
> Thank you for your review! To first address your major points:
>
> 1. We limited ourselves to MNIST and CIFAR10 because this is what the main findings of Humayun et al. 2024 were based on.
> 2. Our paper is indeed mostly a response to Humayun et al.’s findings. As you stated yourself: “Publishing this paper could potentially fix incorrect statements in existing literature”. However we also think we create awareness of the unintended side-effects that can occur during overtraining. We will look to make this more clear.
> 3. We would like to point out that Humayun et al. used MSE loss in their MNIST-MLP experiments and CE loss in their CIFAR10-ResNet experiments. This is not entirely clear from their paper, but a) our reproductions show high similarity to their results , and b) we confirmed this with them. We indeed show that there is no delayed robustness with stronger attacks in figures 5 and 7 for MNIST and CIFAR10 respectively, unless there is something we misunderstand about your comment.
>
> To address your minor points:
>
> 1. Indeed, the experiments of Section 4.1 titled “Mean Squared Error” were trained using Mean Squared Error Loss, as also stated in the description of Figure 2. We will add “Loss” to the title of the section.
> 2. We refer to what is pronounced as “Ortho-Grad”, an optimizer discussed by Prieto et al. 2025. In their paper they also write “⊥Grad”. We will make this more clear.
>
> Now to answer your main question:
>
> We include experiments with both MSE and cross-entropy to match with other works. In particular, our main goal is to critically analyze the results from Humayun et al. 2024. Since they use different losses (MSE or CE) between settings and saw delayed robustness either way, we opted to apply both in all our experiments. The use of MSE definitely still relates to the problem of gradient scaling in Adam, which is why, when we instead use AMSGrad with MSE to solve this issue, we see a lack of delayed robustness. MSE loss has also been used for classification in other papers on grokking, for example Liu et al. 2023.

---

### Official Review · Reviewer_uJgs · 2025-11-01

**Soundness:** 2
**Presentation:** 2
**Contribution:** 2
**Rating:** 4
**Confidence:** 4

**Summary:**

The paper argues that the “delayed robustness” reported by Humayun et al. (2024)—i.e., a sudden rise in adversarial (PGD) accuracy well after overfitting—does not reflect true robustness or grokking. Instead, it is largely an artifact of (i) softmax collapse under cross‑entropy (CE), which makes loss gradients uninformative for PGD, and (ii) Adam’s gradient‑scaling pathology that inflates the effective learning rate when running‑average second moments get tiny; both effects emerge late in training and can mislead PGD‑style attacks. The authors reproduce delayed robustness on MNIST MLPs and CIFAR‑10 ResNet‑18s, then (a) apply stronger or numerically stabilized attacks (AutoAttack+ or a proposed ScaledPGD) and (b) introduce stabilizing interventions (larger Adam δ or AMSGrad; higher‑precision loss/gradients), showing that the PGD gains largely vanish. They further connect the phenomenon to dying neurons, reduced local complexity, and the optimizer “slingshot” effect.

**Strengths:**

1- Clear negative result against a popular hypothesis: The paper demonstrates that the late PGD gains do not persist under stronger or stabilized attacks (AutoAttack+ / Auto‑PGD; ScaledPGD), undermining the interpretation of delayed robustness as “grokking‑induced robust partitions.” The side‑by‑side PGD vs AA+/ScaledPGD gap in Figure 1 and Figure 5 is compelling.

2- Mechanistic decomposition. Two concrete mechanisms are analyzed:
A) Adam gradient scaling: equations (2)–(7) explain how small vt inflates the effective step size, making training non‑convergent late and causing neuron death and local‑complexity drops; AMSGrad/δ↑ mitigations support the causal story (Figure 2, 3, 4).

B) Softmax collapse: a careful numerical account (underflow vs absorption) using the stable logsumexp form (eq. 8–12) and explicit collapse counters that align with loss spikes and PGD rises (Figure 6). The ScaledPGD proposal (eq. 11–12) is a practical attacker‑side fix.

3) Causal interventions rather than correlations. Changing the optimizer (AMSGrad or Adam δ), numerical precision (32→64 bit), and the attack (ScaledPGD/AA+) moves the phenomenon in the predicted direction. This triangulation is well‑executed. Figures 2, 5, 7.

4) Useful diagnostics and instrumentation. The paper quantifies dead neurons, local complexity near data, and collapse counts through training (Figures 3–4, 6, 8), and shows slingshot‑like loss spikes in linear‑time plots (Figures 18–19), producing a coherent diagnostic suite other groups can reuse

**Weaknesses:**

1- Attack coverage and settings are narrow in places.
Protocol rigidity. PGD always uses 10 steps and a fixed step size (0.0156) at ε=0.06 with [0,1] clipping (Appendix A.1). While the paper’s thesis is about PGD’s gradient informativeness, a broader sweep (steps, step‑size schedule, random restarts, ε grid) is needed to fully rule out “insufficient PGD tuning” as the sole culprit in some setups.

AutoAttack usage. In several experiments only AA+’s apgd‑ce component is used; other components (e.g., FAB, Square) are omitted, which could matter especially around gradient obfuscation. This is disclosed but weakens the universal claim.

2- Causality vs correlation for the optimizer story. The Adam‑induced effective step‑size blow‑up is plausible, and AMSGrad/δ↑ help (Figure 2), but there is no direct measurement of the problematic scale factor (e.g., histograms of $\gamma / \sqrt{v_t+\delta}$ over layers/time). Without those, the link from scaling → slingshot → dead neurons → PGD failure is still partly inferential.

3- 64‑bit precision as a “fix” is impractical and only delays problems in ResNets. Double precision removes MNIST collapse (Figure 5), yet for ResNet‑18 it mostly delays the onset (Figure 8). Since 64‑bit training is rarely used at scale, the paper should emphasize optimizer/loss‑level fixes over precision and demonstrate collapse‑free training with realistic toolchains.

4- Limited normative space and datasets. All attacks are ℓ∞ with ε centered at 0.06; there is no exploration of ℓ2/ℓ1 nor datasets beyond MNIST/CIFAR‑10. The claim that “delayed robustness is not grokking” would be stronger with at least one non‑vision or larger‑scale dataset. Figure 10 varies ε only within MNIST.

5- Softmax‑collapse detection could be quantified more tightly. We see the fraction of train samples with collapse (Fig. 6), but not the joint statistics that link collapse severity (max‑min logit gap, counts) to PGD failure per‑example and to AA+/ScaledPGD success. That mapping would directly substantiate the mechanism.

6- Interplay between the two mechanisms is not fully isolated. The paper treats MSE/Adam to diagnose optimizer effects and CE to diagnose softmax collapse. It would be helpful to cross the factors—e.g., CE + AMSGrad/δ↑ and MSE in 64‑bit—to demonstrate additivity or independence. (Some CE 64‑bit results exist, but optimizer cross‑checks for CE are missing.)

7- Attack–defense diagnostics for gradient obfuscation are incomplete. The community “red flags” (transfer attacks, black‑box vs white‑box gap, stepsize sensitivity, EOT for randomness) could be run systematically to show that the late‑phase PGD robustness satisfies classic obfuscation patterns. The current evidence strongly suggests it, but the standard checklist would close the loop

**Questions:**

1- PGD sensitivity: If you vary PGD steps (e.g., 10→50→200) and step sizes (fixed / backtracking), how do late‑phase adversarial accuracies change on MNIST MLP and ResNet‑18? Please plot per‑example attack success vs. step schedule.

2- Random restarts: With ≥20 PGD restarts, does the late‑phase PGD robustness persist? Report mean/variance over restarts.

3- AA+ completeness: For experiments where you used only apgd‑ce, what changes when the full AA+/AutoAttack suite (including FAB and Square) is enabled? Provide both curves and final numbers

4- Transfer & black‑box: Does black‑box transfer from independently trained surrogates attack the late‑phase models better than white‑box PGD? This would be another obfuscation indicator.

5- Other norms: Do the conclusions hold for ℓ2 and ℓ1 (with matched perceptual budgets)? Include ε‑sweeps like Fig. 10 for those norms.

6- Direct measurement: Please report the distribution over iterations/layers of the effective scale factor $\gamma / \sqrt{v_t+\delta}$ and of $v_t$ correlate spikes with loss slingshots, dead‑neuron counts, and PGD failure.

7- Hyperparameter sweep: Beyond δ, how do β₂, β₁, and base LR affect the onset of delayed PGD robustness? Include AdamW and Adagrad/Lion comparisons.

8- Causal intervention: If you clamp $\gamma / \sqrt{v_t+\delta}$ to a max value during late training, do slingshots and dead neurons disappear without otherwise changing the optimizer?

9- Per‑example linkage: For CE models, can you show a scatter of (max‑min logit gap, collapse type/count) vs. PGD success and vs. ScaledPGD/AA+ success at the same points in training (e.g., before/after the PGD jump in Fig. 5)?

10- Optimizer control for CE: Does AMSGrad or δ↑ affect the frequency of absorption collapse (not just underflow)? If not, this would reinforce optimizer‑independence of collapse.

11- Loss‑side fixes: Beyond precision, how do StableMax or logit‑clipping during loss/grad computation compare to ScaledPGD in both attack success and training stability? (The paper cites these ideas in discussion but does not evaluate them.)

12- EOT for numerics: If you inject small logit noise and run Expectation‑over‑Transformation attacks, do PGD results align with ScaledPGD—i.e., does EOT “unstick” PGD when collapse makes the gradient locally flat?

---

> ### Author Response · Authors · 2025-11-28
>
> Dear reviewer,
>
> Thank you for your highly detailed review. To address your questions:
>
> 1. Preliminary results from the start of the project showed that increasing the steps only very marginally decreased the adversarial accuracy in ‘delayed robustness’ regimes. We therefore stuck with 10 steps, because it was fast and still served well as a reproduction of the results by Humayun et al. (2024). We will make this more clear in the Appendix. We also fixed our step sizes to reproduce Humayun rather than e.g. making the step size dependent on the epsilon. See also our next reply.
> 2. Our goal with PGD was solely to reproduce the results of Humayun et al. We therefore do not see the need of optimizing PGD, other than it could serve as another proof that delayed robustness does not exist. Instead we just went straight to Auto-Attack as a stronger attack, rather than optimize PGD.
> 3. Since APGD-CE is very early in the AA+ attack chain and already leaves near 0 adversarial accuracy in the experiments where we use it in isolation, adding the other attacks from the AA+ suite will not further improve our results in the overtraining regime. We do however admit that this is not clear from the text and this might be confusing. To avoid this confusion, we will repeat our experiment with the complete Auto-Attack+ and report those results instead.
> 4. In preliminary experiments we found that models from earlier points in training could be used as a surrogate to attack the late, seemingly robust, models. We did not include these results in the paper, because we believe that the other results are clear to the point that we do not need ‘another’ indicator of obfuscation.
> 5. We did not try L2-norm PGD, as we tried to follow Humayun et al (2024) and they restricted themselves to inf-norm attacks as well. We do not expect the results with L2-norm attacks to be substantially different.
> 6. We saw that most of the dead neurons accumulated in the earlier layers. We believe the distinction between layers is irrelevant for our conclusions however, and the current presentation makes it easier to compare dead neuron introduction between experimental settings.
> 7. For our case, we believe it is sufficient to remove the loss instabilities/slingshots with δ or AMSGrad and show a lack of DR under otherwise exact same settings like in Fig 2. The main result is the accumulation of dead neurons explaining DR.
> 8. Since the only non-fixed parameter in the factor is v_t, this would imply pre-setting global maxima for v_t rather than having AMSGrad infer it on a parameter-wise basis. Since these maxima can be parameter dependent, we think such an experiment is too complex to set up and tune given the marginal benefit it would make for our overall point.
> 9. Could you explain your motivation behind this question? We are not sure what such a plot would prove. ScaledPGD already makes logit gaps irrelevant, and these logit gaps are what is directly linked to the collapse.
> 10. In results that did not make it into the paper we found that, as expected, using AMSGrad, slowed down logit growth and thereby the onset of delayed robustness because of its lower effective learning rate. To also immediately address weakness nr 6: we also tried increasing precision in MSE but there was no difference. From your review we however learn that these findings should make it back to the paper, so we will add these cross-checks back into the appendix.
> 11. We can include stablemax to both our attack- and training regimens but we do not see any real benefit for our purpose. It would just remove numerical stability issues in SCE similarly to our current interventions. Regarding logit clipping, if the definition of Wei 2023 is to be used, it is nearly equivalent to our ScaledPGD-approach which then uses a pseudo-norm instead of the real norm. We find this difference negligible as we are only concerned with getting logits back into numerically stable range, as PGD will take the sign of gradients anyways.
> 12. This could probably work, as long as the logit noise must be relative to the logit magnitudes and capable of bringing the logit differences back to numerically stable ranges. We also do not see the added benefit for the point of our paper. ScaledPGD already deflates the delayed robustness claim as shown.

---

### Official Review · Reviewer_48FM · 2025-11-01

**Soundness:** 3
**Presentation:** 2
**Contribution:** 1
**Rating:** 2
**Confidence:** 3

**Summary:**

The paper studies delayed robustness of models that are trained beyond overfitting, and shows that, in contrary to previous literature, robustness does not obviously increase with increasing training time. The authors then explain the observed phenomenon and show by well crafted experiments, that it is linked to side effects in overtraining, which correlates with observations in the literature. They also confirm that PGD cannot be seen as an absolute metric for robustness, especially in settings where gradients become unreliable, as it may be the case in the observed settings.

**Strengths:**

Extensive experiments are proposed to study delayed robustness, confirm its existence, and explain the underlying phenomenon in overtraining. The experiments are generally well chosen, and quite clearly presented, and offer confirm observations of other papers, as well as offer new insights on the training dynamics and robustness performance in the overtraining regime.

**Weaknesses:**

While the paper is quite extensive in experiments, and (sometimes new) insights on the overtraining regime, the main message of the paper is not really clear. What is the main motivation of studying delayed robustness? What would the results and insights in the paper, bring in terms of constructive solutions for further developments of models that have both good generalisation performance, and robustness?

The structure of the paper, and the definition of the problem should probably be strengthened, so that the reader is properly guided through the numerous experiments, and so that the actual contribution becomes clearer. That would certainly help valorise the extensive experimental work, and the expertise offered by the authors in this paper.

In addition, it may be important to clearly clarify the new insights offered in the paper, with respect to results that conform obsverations proposed in other paper. In the current version, things are a bit inter-twined, which makes it hard to truly appreciate the proposed contribution.

It is also known that PGD is not a perfect proxy for robustness, due to its sensitivity to gradient artifacts. Also, MNIST, that is used in the majority of experiments, is known for being a very poor proxy for developing strong insights on large-scale model training (same for CIFAR10). In order to validate general results and insights, it is important to confirm experimental observations on larger datasets.

Probably, a stronger explanation, or well posed intuitions, about the existence, or the possibility for increased robustness in overtraining regime, would be beneficial for the community. It does not seem to be related to grokking, according to the paper development, but that path is only discussed very superficially, unfortunately. Maybe there could be solid arguments to get inspiration from, in the development of Humayun 2024, that could be transposed to delayed robustness analysis and its connection to grokking, if any?

Formally, eq (1) is not fully correct, as it is missing a connection to the correct label y.

**Questions:**

see above

---

> ### Author Response · Authors · 2025-11-28
>
> Dear reviewer,
>
> Thank you for your review!
>
> To answer your questions:
> 1. Our main motivation for studying delayed robustness was skepticism towards the original Delayed Robustness paper (Humayun, 2024). Our main contribution is to rectify any misconceptions about delayed robustness and create awareness of unintended side-effects during long-term training. We understand that this message may not have been clear and we will look into restructuring or rephrasing, so that this is explicitly mentioned in the paper.
>
> 2. Regarding our choice for PGD, MNIST and CIFAR10: we followed the original paper’s (Humayun et al., 2024) findings. Their main results were also inferred from MNIST and CIFAR10 experiments.
>
> 3. The only link we see between grokking and DR is the occurrence of loss slingshots. These slingshots seem to induce the adversarial accuracy spikes to PGD, by introducing dead neurons (figures 1 and 3). In the discussion we mention the possibility that these dead neurons also simplify the model, and pose it as a possibility for future work to investigate if this is why such slingshots can also induce sudden generalization/grokking in other works. We again will look into restructuring and rephrasing to make this more clear.
>
> 4. Above Eq(1) we say that this is only defined when the sample is classified correctly, so when $f(x) = y$. We will update the formula to make this more clear.

---

### Author Response · Authors · 2025-11-20
**General Statement**

First of all, thank you all for your reviews. There is valuable feedback that we will work on. We hope to come back to each of you with individual responses soon. Given the nature of the reviews, however, we felt it necessary to make a general comment about the purpose of our paper.

Our paper shows that the delayed robustness phenomenon is not grokking, but is either false robustness or an artifact of unintended training behaviors. This finding is indeed a direct refutation of the paper of Humayun et al. (2024), as some of the reviews have correctly noted. This ICML paper has been widely cited in subsequent literature as an example of grokking.

Our refutation of the claim of Humayun et al. is based on the experimental setting of the original paper. Therefore,  in the spirit of the self-correction of science, we do not understand why reviewers want us to extend our findings to larger datasets or why they have doubts about our work providing a worthwhile contribution.

We may not have communicated our goal and scope clearly enough, and we will improve our paper in that respect. Under the assumption that we clarify our paper satisfactorily, we ask the reviewers if they still believe our contribution is not up to par for a presentation at ICLR.

---

### Meta-Review · Area_Chair_R21v · 2025-12-24

**Summary:**

The reviewers had the following concerns:
1. Limited contribution. The main purpose of this paper is to provide series of empirical experiments to refute claims from the paper by Humayun et al.; it is unclear whether there is any new insights from this practice. Moreover, the writing clarity needs improvement to articulate the main message of this paper.
2. Insufficient experiments. This issue seems to be two-fold: Firstly, the experiments are with small models and simple datasets; whether the insights offered in the paper can be said for large-scale problems remain unclear. Second, the paper compares with several algorithmic variations to isolate the reasons behind the delayed robustness, but those experimental designs can be improved.

**Reviewer Concerns:**

The author rebuttal is relatively short, the authors did not provide any revision to the paper. I believe both concerns are still outstanding.

**Reviewer Scores:**

I think all the reviewers would keep their scores (4, 4, 2, and 2).

---

### Decision · Program_Chairs · 2026-01-26

Reject